# Focal Minimally Invasive Treatment in Localized Prostate Cancer: Comprehensive Review of Different Possible Strategies

**DOI:** 10.3390/cancers16040765

**Published:** 2024-02-13

**Authors:** Eliodoro Faiella, Domiziana Santucci, Giulia D’Amone, Vincenzo Cirimele, Daniele Vertulli, Amalia Bruno, Bruno Beomonte Zobel, Rosario Francesco Grasso

**Affiliations:** Department of Radiology and Interventional Radiology, Fondazione Policlinico Universitario Campus Bio-Medico, Via Alvaro del Portillo, 00128 Rome, Italy; e.faiella@policlinicocampus.it (E.F.); d.santucci@policlinicocampus.it (D.S.); v.cirimele@policlinicocampus.it (V.C.); daniele.vertulli@unicampus.it (D.V.); amalia.bruno@unicampus.it (A.B.); b.zobel@policlinicocampus.it (B.B.Z.); r.grasso@policlinicocampus.it (R.F.G.)

**Keywords:** focal therapy, ablation techniques, prostate cancer, cryotherapy, irreversible electroporation, microwave ablation

## Abstract

**Simple Summary:**

This systematic review includes 23 studies about minimally invasive methods for selective treatment of localized prostate cancer. At present, standard treatment options for localized prostate cancer are active surveillance and whole-gland treatment (radical prostatectomy and radiotherapy). Regrettably, morbidities are represented by urinary incontinence, erectile dysfunction, and bowel dysfunction, which can worsen the quality of life without necessarily improving the patient’s oncological outcome. Minimally invasive methods have demonstrated encouraging results in terms of functional outcomes and low adverse events. The focus of this review is to update evidence about the oncological effectiveness of three minimally invasive techniques (cryoablation, irreversible electroporation, and microwave ablation).

**Abstract:**

Background: Focal therapy is a promising, minimally invasive method for the treatment of patients with localized prostate cancer. According to the existing literature, there is growing evidence for positive functional outcomes and oncological effectiveness. The aim of this review is to evaluate the technical efficacy of three minimally invasive techniques (cryoablation, electroporation, and microwave ablation) and their impact on quality of life in patients with prostate cancer. Methods: Studies between January 2020 and July 2023 were selected using PubMed, Embase, and The Cochrane Library and analyzed following PRISMA guidelines; they have not been registered. Results: Twenty-three studies investigating three different sources of energy to deliver focal therapy were found. Thirteen studies evaluated the performance of the cryoablation therapy, seven studies of the irreversible electroporation, and three studies of microwave ablation option. The majority of studies were retrospective cohort studies. Cryoablation showed excellent oncological outcomes for low-grade prostate cancer, whether performed on the lesion, on the hemigland, or on the entire gland, with the best results obtained for patients with intermediate risk. Irreversible electroporation showed promising oncological outcomes with no significant changes in functional outcomes. Microwave ablation showed great early functional outcomes. Conclusions: The oncological effectiveness of minimally invasive treatment in comparison to standard of care is still under investigation, despite encouraging results in terms of functional outcomes improvement and adverse events reduction. More comprehensive research is needed to fully understand the function of minimally invasive treatment in patients with localized PCa.

## 1. Introduction

At present, standard treatment options for localized prostate cancer (PCa) are active surveillance (AS) and whole-gland treatment, represented by radical prostatectomy (RP) and radiotherapy) [1]. Regrettably, morbidity weaknesses of these last two are urinary incontinence, erectile dysfunction, and bowel dysfunction, which can worsen the quality of life and not necessarily improve the patient’s oncological outcome [2,3]. Even though AS was linked to a higher rate of disease progression and metastases, earlier research has already shown how the 10-year cancer-specific survival for patients with low- and intermediate-risk prostate cancer treated with radical prostectomy and external radiotherapy is comparable to that of AS [4]. To improve the benefit-to-risk ratio, alternative medicines have been studied to minimize side effects while preserving a favorable oncological result. Focal treatment (FT) seems to be one such promising option among them. In order to preserve nearby vital structures, FT attempts to treat the “index lesion”. Tumor size, position, and grade may determine the patient’s prognosis by influencing the probability of metastasis [5]. KJ Tay et al., at the International Delphi Consensus meeting, concluded that AS should be prioritized in males with low-risk illness because there is no net benefit from FT, whereas FT should be explored in individuals with intermediate PCa risk [6]. Over the past few decades, numerous energy source types in FT have been reexamined. These consist of cryotherapy, photodynamic treatment, radiofrequency ablation (RFA), irreversible electroporation (IRE), high-intensity focused ultrasound (HIFU), focal brachytherapy, and focal laser ablation (FLA). The goal of this updated systematic review is to evaluate recent findings on functional and oncological outcomes of FT for patients with localized PCa and disease recurrence. Based on the numbers of studies present in the medical literature, we chose to discuss two FTs with a high number of cases, cryotherapy and irreversible electroporation, and one FT with fewer cases, but is still very promising, microwave ablation.

## 2. Materials and Methods

This systematic review was carried out according to the Preferred Reporting Items for Systematic Reviews and Meta-analyses (PRISMA) guidelines [7] and has not been registered. The included studies and exclusion adopted criteria are shown in Figure 1.

The terms “prostate cancer”, “focal therapy”, “ablation techniques”, and the names of the energy sources were searched in the databases of PubMed (Medline), Embase, and The Cochrane Library. Since Valerio et al. [8] had already conducted a search up to October 2015, we searched for studies published from January 2019 to July 2023. Studies were considered if they included FT as the main therapy for primary tumor or disease recurrence and one of these two endpoints: functional outcome (for example, impotence and incontinence) and oncological outcome (for instance, post-procedural biopsy, prostate-specific antigen (PSA), and disease-free survival). Single-arm studies, retrospective and prospective cohort studies, randomized controlled trials (RCTs), systematic reviews, and meta-analyses were included. Case studies, review articles, and abstracts from congresses were not included.

## 3. Results

Twenty-three studies between January 2019 and July 2023 were selected. Of them, 13 studies performed cryoablation, 7 irreversible electroporation procedure, and 3 microwave ablation. The majority of studies were retrospective cohort studies (Table 1).

### 3.1. Cryoablation (CRA)

Tan WP et al. [9] included in their study 260 men with primary PCa that were treated with primary whole-gland cryoablation (WGC) of the prostate. Biochemical recurrence-free survival (BRFS) at 10 years was 84%, failure-free survival (FSS) was 66%, and metastasis-free survival (MFS) was 96%. Following CRA, both the American Urological Association’s symptoms score and bother index remained unchanged. Pre- and post-cryoablation median International Index of Erectile Function scores were 7 and 1, respectively. Only five patients (2%) had stress urinary incontinence and no patient had fistula formation. Six (2.3%) patients experienced Clavien–Dindo adverse events of grade > 2 [9]. With this study, the authors continue the cryoablation evaluation started by their previous study, still directed by Tan WP [10], in which, among 82 men, 11 men underwent salvage partial gland (group A) ablation and 71 men underwent primary customized partial gland cryoablation (group B). In group A, failure-free survival was 98%, 89%, 84%, 75%, and 75% at 1 to 5 years. In group B, failure-free survival was 100%, 80%, and 40% at 1 to 3 years, respectively. At three months and for the duration of the follow-up period, all 71 patients in the primary therapy group were free of pads. The International Index of Erectile Function (IIEF) scores of the men treated with primary subtotal CRA and primary hemicryoablation were lower after treatment than those of the men treated with primary focal CRA. The American Urological Association (AUA) symptom scores decreased regardless of the partial gland ablation technique employed; subtotal ablation had the lowest score when compared to hemiablation and localized cryoablation. In group A, no patients developed fistulas, and in group B, one (9%) patient did [10]. Mendez M.H. et al. [32] in 2015 compared whole-gland (WG) CRA with localized ablation, revealing a significant difference in favor of focal therapy over WG CRA.

Bossier R et al. [12] in their 2023 study found no difference in pad-free continence between the two groups, hemi- and whole-gland cryoablation (both 83%), and could not confirm that those who received focal treatment had better erectile function preservation (after 24 months, 68.8% vs. 46.8% ESI, *p* = 0.001).

All patients in Chuang R. et al.’s prospective observational experiment on hemigland cryoablation as a primary treatment for unilateral prostate cancer (csPCa) had csPCa (GG2 or higher), and outcomes were uniformly assessed by thorough MRGB (MRI-guided biopsy) at baseline and at short- and intermediate-term follow-up. CsPCa was found to be gone in 82% of men at follow-up MRGB six months following treatment. When MRGB was repeated 18 months after treatment, the 82% efficacy was still present and only 1 of 27 men exhibited contralateral csPCa [11]. Others have reported a drop in serum PSA levels following a successful treatment, but the present group’s PSA decreases were not statistically significant [33].

Gregg JR et al. [20] conducted a prospective nonrandomized controlled trial to assess the effectiveness of subtotal prostate ablation in a subset of men with grade group (GG) 1–2 Pca at baseline and confirmed by biopsy. The ipsilateral hemigland and contralateral anterior prostate were both subjected to “hockey-stick” CRA. Following localized ablation, prostate biopsies and quality of life (QoL) tests were performed at 6, 18, and 36 months. At first control, 12/23 (52%) patients were disease-free, and every patient had kept urine control without the need for pads due to incontinence. At 3 and 6 months, there was a substantial sexual function reduction (*p* < 0.01 for both), but at following time points, an improvement of this was observed.

Khan A. et al. included 163 patients in their retrospective study and reported disease-free survival rates of 78%, 74%, and 55% for low-, intermediate-, and high-grade malignancies, respectively, after CRA, contributing to the majority of evidence about treatment outcomes on high-grade prostate malignancies. Jones JS and Rewcastle JC, in 2008, had already focused on the effects of primary CRA on patients with Gleason 8, 9, or 10 localized or high-grade Pca, reporting good biochemical disease-free survival at five years (64.4% by ASTRO criteria) [13].

In Selvaggio O. et al.’s retrospective study, 110 patients, of which 54 (49.1%) had low-risk, 42 (38.1%) intermediate, and 14 (12.8%) high-risk Pca, underwent PGC (partial gland cryoablation). A biochemical recurrence survival (BCS) and treatment-free survival (TFS) of 75 and 81% were recorded at a median follow-up of 36 months. Five years later, CRS (clinical recurrence) was 71.5% and BCS was 68.5%. When compared to the low-risk group, high-risk prostate cancer was linked to lower TFS and BCS curves (all *p* values were <0.03) [14].

With regard to the role of MRI in detecting relapse after CRA, interesting insights were provided by Wysock et al.’s study [15]. A total of 132 men underwent follow-up for at least 24 months. In 12 men, biopsies revealed clinically significant PCa. At 36 months, in-field, out-of-field, and overall clinically significant cancer recurrence-free rates were estimated by the model to be 97% (95% CI: 92–100), 87% (95% CI: 80–94), and 86% (95% CI: 78–93), respectively. At 36 months, the model’s estimated freedom from failure proportion was 97% (95% CI: 93–100).

Patients with a PIRADS 4 or 5 single lesion and a worrisome prostatic specific antigen (PSA) value were enrolled in Misuraca L. et al.’s research [21] to undergo transperineal 3D MRI–US guided PB and TRUS-guided localized CRA. Focal CRA was carried out following the confirmation of PCa in frozen portions. The index lesion in all patients had been totally eradicated, according to MRI scans, and mean PSA values had decreased from the baseline of 12.54 to 1.73 ng/mL at the three-month evaluation. All patients’ urinary potency and continence were retained. One patient underwent a new, similar procedure after having a suspicious ipsilateral recurrence on MRI at the 1-year follow-up. Post-follow-up was uneventful and all patients’ PSA levels stayed stable.

In their study, Tan W.P. et al. [17] examined the functional and oncological results for males treated for radiation-resistant/recurrent prostate cancer (RRPC) with salvage whole-gland cryoablation (SWGC). The study included 110 men with biopsy-proven RRPC. Patients without biochemical recurrence (BCR) after SWGC had a median follow-up of 71 months. BRFS (biochemical recurrence-free survival) was 81% after two years and 71% after five. There was a negative correlation between worse BRFS and a higher PSA nadir after SWGC. Prior to SWGC, the median International Index of Erectile Function-5 score was 5, and after SWGC it was 1. At 3 months and 12 months, respectively, stress urinary incontinence, defined strictly as the use of any pads after treatment, was 5% and 9%. Three patients (2.7%) experienced adverse events of Clavien–Dindo grade 3.

After radiation therapy failed, Nair S.M. et al. [18] attempted to add salvage local treatment for PCa patients. Two propensity score-matched analyses were performed: (1) no salvage therapy, 196 NST vs. 98 salvage cryotherapy (sCT); and (2) salvage high focal ultrasound (sHIFU), 177 NST vs. 59. In the first comparison, sCT was responsible for 80 fatalities and prostate cancer for 24, while NST was responsible for 49 deaths and PCa for 78. In the second comparison, 52 deaths—31 from prostate cancer—were associated with NST, compared to 18—9 associated with sHIFU. Due to the smaller sample size and shorter follow-up of the sHIFU cohort, there were no appreciable differences in cancer-specific survival (CSS) or overall survival (OS).

Included in this research on metastatic PCa CRA is a study by Smigelski M. et al. [16]. Following a mean follow-up of 39 months, a total of 187 patients with locally recurrent PCa following radiation underwent salvage CRA of the prostate for analysis. The serum PSA level at the time of CRA was a significant predictor of BR on both univariate and multivariate analyses (*p* = 0.001). Patients with a pre-cryoablation PSA of less than 4 ng/mL had a 5- and 8-year BRFS of 56% and 37%, respectively. Over a period of 5 and 8 years, respectively, patients with pre-cryoablation PSA levels of 10 ng/mL or greater experienced BRFS rates of just 1% and 7%. Patients who had pre-cryoablation PSA levels between 4 and 9.99 ng/mL had a mixed prognosis for survival. At a mean of 31 months after surgery, 32% of the patients began hormonal therapy for disease progression; 97% and 92%, respectively, of people survived for 5 and 8 years overall.

In two groups consisting of 54 patients with newly diagnosed metastatic prostate cancer (mPCa) receiving CRA with androgen deprivation therapy (ADT) or ADT alone, Wang N. et al. [19] examined the oncological outcomes and therapeutic value in symptom control. The median follow-up period for groups A and B was 40 (27–53) months and 39 (31–54) months, respectively. Patients in group A had a lower median PSA nadir (0.025 ng/mL vs. 0.230 ng/mL, *p* = 0.001), a shorter median metastatic castration-resistant prostate cancer (mCRPC)-free survival (39 months vs. 21 months, *p* = 0.007), and a longer median failure-free survival (FFS) (39 months vs. 21 months, *p* = 0.005). There was no difference in overall survival or cancer-specific survival between the two groups.

### 3.2. Irreversible Electroporation (IRE)

Blazevski A. et al. [34] assessed focal irreversible electroporation (IRE), which can be used to treat men with localized PCa. In this study, 123 patients underwent IRE as part of their focal therapy. In 90.2–97.3% of patients, the follow-up biopsy revealed no evidence of persistent disease. At three years, 96.75% of patients avoided a whole-gland surgery. Patient age ranged from 62 to 73 years. PSA levels prior to surgery ranged from 3.7 to 8.0 ng/mL. In-field recurrence was present in 2.7–9.8% of patients on post-treatment TTMB (transperineal template mapping biopsy). FFS at 3 years was 96.75%, metastasis-free survival at that time was 99%, and overall survival was 100%. Eighteen patients in total required salvage therapy (12 underwent repeat IRE, and 6 underwent whole-gland therapy). Six months after treatment, the mpMRI had a negative predictive value of 94% and a sensitivity of 40% for detecting in-field residual disease. A total of 40/53 (76%) of the patients who completed questionnaires had no change in erectile function, and 80/81 (98.8%) of the patients continued to be pad-free.

Data on 229 patients with locally advanced clinically relevant PCa were presented by Scheltema et al. [23] in an attempt to assess the longer-term oncological and functional results of focal IRE as the primary treatment, with a median follow-up of 60 months. The median interquartile range PSA level was 5.9 (4.1–8.2) ng/mL, the median age was 68 (64–74), and 86% of the participants had intermediate-risk diseases and 7% had high-risk diseases. Thirty-eight patients in total (17%) advanced to radical therapy at a median (IQR) of 35 (17–53) months after IRE. At three years, Kaplan–Meier FFS rates were 91%, at five years, 84%, and at eight years, 69%. PCa-specific and OS were both 100% (229/229), while metastasis-free survival was 99.6% (228/229). The following biopsy revealed residual csPCa in 24% (45/190) of the cases, and MRI revealed complete ablation in 82% (186/226) of the cases. Short-term urinary continence was maintained (98%, 3 of 144 at baseline, 99%, 1 of 131 at 12 months), but the number of erections strong enough for sexual activity fell by 13% from baseline (71% to 58%).

de la Rosette J. et al. [24] conducted a multicenter, randomized, single-blind, 2-arm intervention comparing focal and extended IRE in men with localized low–intermediate-risk PCa. Focused and extended IRE were performed on a total of 51 and 55 individuals, respectively. A 30-month follow-up period was the median. At three months, the two groups’ rates of erectile dysfunction and adverse events were comparable. At three months, the focal ablation group appeared to have higher International Index of Erectile Function scores. It also appeared to have higher Expanded Prostate Cancer Index Composite-sexual function scores over time than the extended ablation group. Other QoL metrics did not significantly differ between the two groups. In the focused and extended IRE groups, the rate of residual clinically relevant Pca (Gleason 3 + 4) at 6 months after prostate biopsy was 18.8% and 13.2%, respectively, without statistically significant differences.

Gielchinsky I. et al. [25] retrospectively studied 45 cases of primary (N = 38) and salvage (N = 7) Pca patients treated with IRE. Prior to therapy, all patients underwent a PET-PSMA scan and transperineal MRI/US fusion biopsy to confirm the single lesion. In the daycare theater, a transperineal Nano-Knife IRE system was used. Following the procedure, patients had mpMRI at 6 months, blood PSA, and a confirmatory biopsy at 1 year. During the first year, life quality was assessed. Age 69 years, starting PSA 5.6 ng/dL, lesion size 0.8 mL, and ISUP Group 2 (1–3) were the median values for the primary subgroup analysis (N = 38). Age 76, starting PSA of 11.9 ng/dL, lesion size of 2.0 mL, and ISUP Group 4 (1–5) were the median values for the salvage subgroup analysis (N = 7). Five (3–7) days is the average length of a catheterization. No problems related to Clavien–Dindo > 1, readmissions, incontinence, strictures, or fistulas were documented. The mpMRI clearance improved by 84%, the in-field lesion dropped by 4%, the out-field new lesion increased by 12%, and the primary group PSA decreased by 39%. Four patients had biopsies at one year that showed clinically significant PC that was out-of-field; as a result, three had re-IRE and one had radiation therapy. After a year, 52% of the salvage subgroup MRIs remained under active observation, with 60% of them showing no abnormalities.

A total of 41 consecutive PCa patients (International Society of Urological Pathology Grade 1–2, PSA ≤ 15 ng/mL, ≤ cT2b) from November 2014 to July 2021 were included in the Miñana López B. et al. [26] study, which had a median follow-up of 36 months. Index lesion ablation was carried out with a safety margin.

A total of 30 patients (73%) had tumors classified as grade 1 by the International Society of Urological Pathology, 10 (24%) as grade 2, and one (2.4%) as grade 3. Recurrence was noted in 16 of the overall cohort’s 41 cases (39%) and 16 of the 33 cases (48.4%) of those who underwent biopsy. Recurrence in the field was found in 5 cases (15%) and outside the field in 11 cases (33.3%). Significant tumors (Gleason pattern 4–5; more than 1 core or any >5 mm involvement) were seen in 10 of 41 (24.6%) cases, including 3 of 5 (60%) with in-field recurrences. The median time without a recurrence was 32 months (95% CI: 6.7–57.2). Twenty-six patients (63.4%) were exempt from receiving salvage therapy.

In their nonrandomized controlled trial with 109 patients, Wang H. et al. [27] reported encouraging efficacy with no impact on functional outcomes for patients undergoing extensive focal ablation with H-FIRE for localized Pca. Out of the 100 patients who underwent biopsy at that time, the 6-month csPCa rate was 6.0%. The 20% threshold for superiority over the historical control was met. Two of the 14 individuals had a Gleason score of 7, and the other 12 had a score of 6. Pca was discovered in these patients. At six months, the median International Prostate Symptom Score was 4.5, the median International Index of Erectile Function 5 score was 2.0, and the median PSA level was 1.08 ng/mL. Another significant finding in the current study supported the minimal damage caused to critical structures (such as blood vessels and nerves) by H-FIRE by showing a significant correlation between higher ablation ratio and lower residual tumor but not IPSS or IIEF-5 score.

More studies have been developed to deepen H-FIRE outcomes knowledge, such as He B.M. et al.’s [35] future multicenter and single-arm objective performance criteria study, in which the aim was to assess the efficacy and safety of the H-FIRE ablation for Pca.

In the study of Kızıl P. G. et al. [28], in addition to the limitation of the small cohort study, the preliminary clinical and mpMRI results after IRE were encouraging. Of the ten patients that received IRE treatment, six patients came in for control visits after completing their six-month follow-up. The average PSA level reduction after six months was 73%. Results from the IIEF (International Index of Erectile Function) were observed to have barely changed. Except for one patient, diffusion restriction was seen to have vanished on mpMRI, and PI-RADS scores had fallen.

### 3.3. Microwave Ablation (MWA)

Oderda M. et al. [29] reported early functional outcomes of targeted microwave ablation (TMA) procedures on low- to intermediate-risk Pca. The median age of the patients was 67 years. The prostate volume was 51 cm3, the median PSA level was 5.4 ng/mL, and the median MRI lesion size was 10 mm. The surgery took a median of forty minutes, with ten patients having stage 2 Pca and one having stage 1 illness. There were no reported intraoperative complications. Not a single patient was released from the hospital with a urinary catheter; all operations were completed as outpatient procedures. Following surgery, no grade ≥ 2 problems were reported. The results showed no discernible changes in the IIEF-5 (*p* = 0.18), IPSS (*p* = 0.39), or PSA (*p* = 0.46). The postoperative Visual Analogic Scale (VAS) score was 0 in all patients at 24 hours.

Some oncological outcomes information came from Delongchamps N.B. et al.’s [30] study. They performed transrectal OBT fusion with targeted FMA ed with an 18-G needle on 10 patients with visible index tumor with Gleason < 3 + 4. Seven days after ablation, the main outcome was an MRI showing total overlap of the index tumor due to ablation zone necrosis. Next to the operation, none of the ten patients experienced pain or rectal bleeding, and they were all released the next day. Eight (80% (95% CI: 55%–100%)) individuals had full necrosis of the index tumor on MRI seven days following ablation.

Chiu P.K. et al. [31] investigated the efficacy of transperineal targeted MWA in treating localized PCa. There were 23 areas being treated in the first 15 men. The median PSA level decreased from 7.7 to 2.4 ng/mL. In the 6-month biopsy for the primary outcome, there was no cancer in 91.3% of the ablated area. One-third of patients with per-patient analysis had in- or out-of-field positive biopsy results at 6 months.

## 4. Discussion

One of these treatment methods that shows the most promise is cryoablation, since it is the least invasive, causes less damage to adjacent tissues, and offers a better preservation of tumor antigens Under local anesthetic, cryoablation is a percutaneous technique that takes 30 to 60 minutes to complete. A cryoprobe is inserted straight into the tumor using an imaging method (ultrasound, CT scan, or magnetic resonance imaging) as guidance. Throughout the process, an ice-ball’s creation is tracked through a sequence of freeze/thaw/freeze cycles that quickly freeze the tumor to the point of necrosis (goal temperature ≤ −40 °C), thus eliminating it. Prostate, liver, kidney, and lung malignancies have all been treated with cryoablation [36].

Electroporation is a technique used to increase the permeability of the cell membrane, with respect to ions and molecules, through the application of short pulses of high electric fields. It can be reversible (RE), if permeabilization is temporary, or irreversible (IRE), when cell death results in irreparable damage to the cell membrane [22,37]. Through the application of pulses of a few hundred volts and lasting the order of a few milliseconds, it is possible to control the effect produced by the electric field on the lipid bilayer, making sure that the radius of the pores that are formed is large enough to allow the passage of the molecules to be injected but also lower than the critical value beyond which the cell is no longer able to return to its initial state (closure of pores). When, on the other hand, the pores do not close, the phenomenon of IRE occurs. Recent advances show the possibility of applying IRE to clinical cancer treatments. The absence of thermal impact of the IRE technology on structures with an adventitia layer is one of its main advantages. As a result, there is little to no impact on the vessels and nerves, which lowers the likelihood of side effects and offers a significant advantage over other focal treatments [38].

One of the most recent advancements in the field of heat therapy is microwave ablation (MWA), which involves delivering microwave radiation precisely while being guided by mpMRI and ultrasound. Due to the quick tissue heat propagation in this technique, thermal tumor ablation is an effective way to kill cancer cells, by raising the temperature above the usual physiological threshold with little to no negative effects on neighboring organs. Although MWA ablation has a long history of efficacy in treating metastases in the kidney, bone, liver, and lung, its application to the treatment of localized Pca is relatively new, as evidenced by the publication of only one other small trial involving ten patients. Comparing the MWA approach to other available targeted treatment modalities, there are a few advantages. For example, it may penetrate deeper and heat a bigger volume than radiofrequency ablation, it is less vulnerable to the “heat sink” effect caused by adjacent capillaries [39], and it has the capacity to penetrate deeper and heat a larger volume [39].

With regard to CRA, several studies suggested that in localized PCa, both the whole-gland and individual partial-gland CRA treatment can produce excellent oncological outcomes [9,10,12,32]; however, the reported results are different in terms of functional outcomes: while Tan W.P. et al. [9,10], in two different papers, demonstrated that the partial option showed slightly better functional outcomes, such as urinary incontinence, both Bossier R. et al. [12] and Mendez M.H. [32] showed no significant differences compared to total-gland CRA.

As we might expect, the hemigland ablation for unilateral PCa also demonstrated excellent effectiveness and safety [33]. In addition, the subtotal (hockey-stick template) CRA of the prostate proposed in one work provided an optimal oncologic control to targeted tissue in a generally low-risk cohort, with minimal impact on sexual and urinary function [20].

Age was not associated with worse outcomes, as Selvaggio O. et al. [14] showed, suggesting that partial-gland CRA could be a useful treatment for low- to intermediate-stage PCa in older patients, where a curative strategy is appropriate in terms of life expectancy and quality of life. Regarding high-grade PCa ablation, although the findings of Khan et al.’s study [13] were promising, outside of clinical studies, clinicians do not advise focal ablation for individuals with high-risk PCa [40]. In fact, according to current AUA/ASTRO guidelines, patients with intermediate-grade cancer should be the main beneficiaries of prostate ablation. EAU guidelines for the elderly recommend a watchful waiting strategy over active treatment if the life expectancy is less than 10 years [1]. Middle-aged patients (75 years or older) may have a >10-year life expectancy but are frequently ineligible for radical treatments. Sometimes they may exhibit even worse side effects as a result of ongoing ADT; for this reason, PCa management in these patients is a hot topic among uro-oncologists.

With regard to the role of multiparametric (mp) MRI in detecting relapse after CRA, interesting insights were provided by Wysock et al.’s study [15]. The low in-field cancer detection rate at 3 years suggested that localized cancers had been successfully ablated. On the other hand, the out-of-field detection rate emphasized the necessity of ongoing surveillance after partial-gland CRA. It appears that mpMRI has a limited function in detecting clinically significant recurrences after two years, as several of the recurrences revealed very little clinically significant illness below the mpMRI detection threshold. In order to determine the optimal timing to perform a biopsy, these findings emphasize the significance of long-term surveillance and the necessity of identifying both early and late predictors of clinically significant Pca recurrences.

In an innovative work, Misuraca L. et al.’s study [21] on concomitant transperineal 3D MRI–US-guided PB and TRUS-guided focal CRA took a step closer to a minimally invasive and patient-tailored approach, showing how it can be possible to diagnose and treat a PCa index lesion in a single session.

CRA has also been proposed as adding salvage local treatment in PCa patients who have failed radiation therapy, with excellent results in terms of CSS and OS, demonstrating how, in selected men with recurrent PCa post-radiation, further local treatment may lead to benefits in CSS [18].

Extensive research on locally spread metastatic PCa CRA has produced some intriguing findings. Specifically, Smigelski M. et al. [16] reported a unique salvage surgical approach that combined robotic excision of the seminal vesicle with CRA of the prostate for locally recurrent PCa of the seminal vesicle with or without prostate involvement following radiation therapy or focused therapy. Their results led to the recommendation that men who experience unilateral seminal vesicle recurrence following primary radiation therapy consider a bilateral salvage-focused CRA and robotic seminal vesiculectomy. If there is no evidence of contralateral disease, then for men who have unilateral seminal vesicle and prostate involvement after primary partial CRA, unilateral salvage focal CRA and seminal vesiculectomy are suggested.

It is also possible to add the CRA as a therapeutic option to the ADT, due to promising results in determining an increase of failure-free survival and metastatic castration-resistant prostate cancer-free survival, relieving urinary symptoms, and decreasing the need for treating primary lesions.

In patients with localized clinically significant PCa, it has been demonstrated by several studies than focal IRE, in particular if performed by highly qualified professionals using a strict algorithm for selection, treatment, and follow-up, is a safe and effective procedure. In detail, IRE of primary lesions demonstrated high rates of oncological control success. Scheltema M.J. et al. [23] demonstrated how, in some men, focal IRE offers adequate not only local, but also distant, oncological control. Although salvage results were not as good as primary results, they frequently provided oncological control, sparing the hormonal therapy [34]. With regard to patients’ QoL, despite the urinary incontinence rate being very low and comparable to those obtained by CRA and lower than radical therapy, a percentage that ranged from 9% to 24% of men treated with IRE exhibited change in erectile function [23,24,25,26,28,34].

In their recent interesting study, Wang H. et al. claimed that the oncologic outcome and functional outcome must be balanced in focal therapy for PCa. Results of this nonrandomized controlled trial indicated that patients receiving extensive focal ablation with H-FIRE for localized PCa experienced encouraging efficacy and little impact on functional outcomes [27].

Only few studies investigated the application of the targeted microwave ablation (TMA) procedure for PCa treatment. However, all available studies that reported early functional outcomes of targeted microwave ablation (TMA) procedure on low- to intermediate-risk PCa concluded that TMA is safe, feasible, and well tolerated [29,30,31]. QoL, erectile function, uroflowmetry, or urinary symptoms evaluated after six months were allayed.

In conclusion, over the past 5 years, only 25 studies have been conducted on cryoablation, irreversible electroporation, and microwave ablation in the treatment of localized prostate cancer.

CRA showed excellent results for low-grade PCa, whether performed on the lesion, on the hemigland, or on the entire gland. However, the best results in terms of clinical outcomes and complications ratio were obtained for patients at intermediate risk. EAU guidelines do not yet indicate CRA treatment for high-risk patients.

Preliminary clinical results for IRE were encouraging. Functional outcomes seemed to not have changed significantly, while oncological outcomes were promising. To establish this new treatment paradigm as a legitimate treatment option, long-term follow-up and external validation of these findings are needed. In particular, future studies comparing H-FIRE directly with a thermal energy platform should be conducted with large sample sizes to also investigate this new technique.

In men with localized PCa, transperineal TMA guided by MRI–ultrasound fusion guidance and organ-based tracking was demonstrated to be productive, secure, and simple to use. However, with all studies being recent (follow-up up to 6 months), no long-term functional outcomes or cancer control outcomes are available.

The main limitation of this review is linked to the variability of the studies included in terms of the sample examined (low number of patients enrolled in the studies) and the results analyzed, leading to a difficult unambiguous interpretation.

Furthermore, in the current literature, there is a lack of prospective randomized and nonrandomized studies that compare these strategies to active surveillance or radical therapy, leading to primordial results which need confirmation and validation in wider samples and multicentric evaluation. However, the studies examined mostly show promising results which indicate the need to continue research in this field of application, particularly in the context of increasingly personalized and minimally invasive precision medicine. In addition, since prostate cancer is a multifocal tumor, it is very difficult to detect all lesions by TRUS (transrectal ultrasound) or MRI.

## 5. Conclusions

The oncological effectiveness of minimally invasive treatment in comparison to standard of care is still under investigation, despite encouraging results in terms of functional outcomes improvement and adverse events reduction. In this direction, more comprehensive research is needed to fully understand the function of minimally invasive treatment in patients with localized PCa.

## Figures and Tables

**Figure 1 cancers-16-00765-f001:**
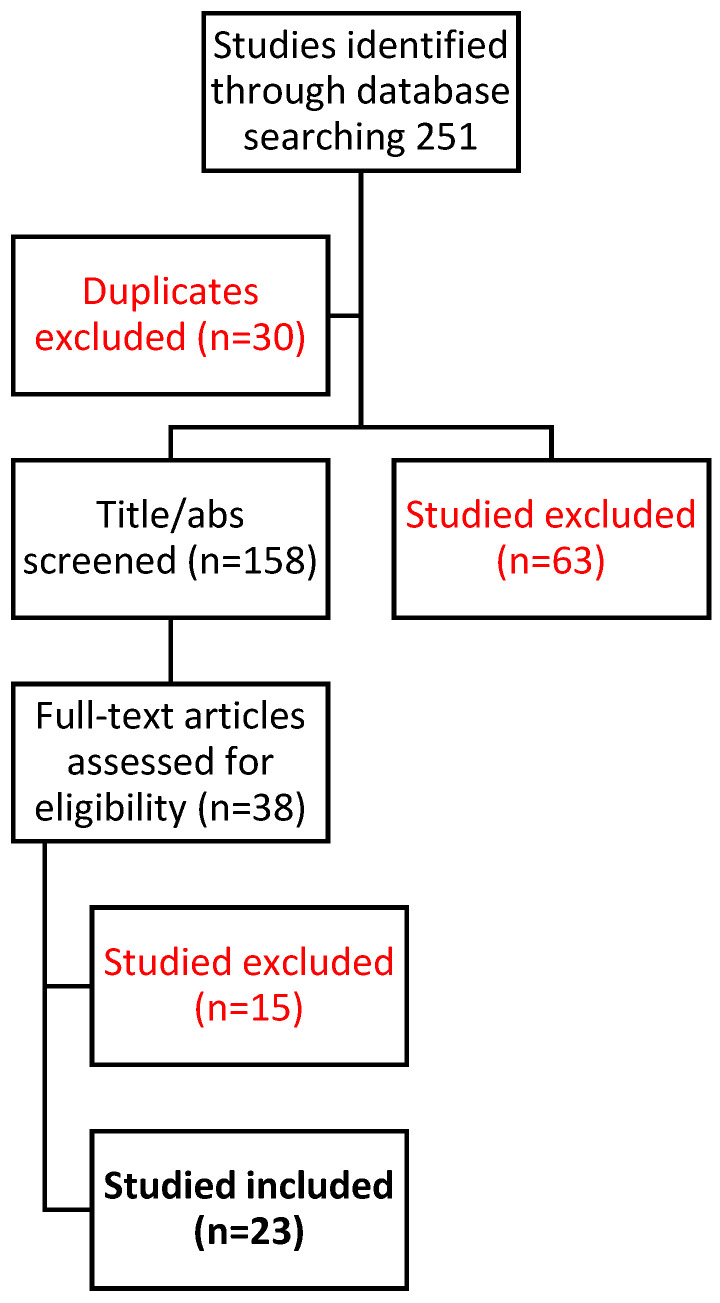
PRISMA flowchart of included studies.

**Table 1 cancers-16-00765-t001:** Summary of the 23 selected studies published from January 2019 to July 2023.

Cryoablation	Study Year	Study Design	Stage/Grading	Type of Ablation	Biopsy	Imaging	Patients (*n*)	Primitive or Recurrence	Functional Outcomes	Oncological Outcomes	Complications	Disease-Free Survival	Overall Survival
[9]	2022	Retrospective cohort study.	N/A.	WGC.	N/A.	N/A.	260	Primitive.	erectile dysfunction (1 post cryoablation vs. 7 before), 2 stress incontinence	BRFS 84%, FFS 66%, MFS 96%.	2.3% Clavien—dindo > 2.	84% BRFS.	N/A.
[10]	2021	Retrospective cohort study.	N/A.	WGC/hemi/focal.	N/A	N/A	82	Both.	no stress incontinence. Erectile dysfunction	FFS 5 years 75% (primary group), 40% (salvage group).	1 fistula in the salvage group.	FFS 5 years 75% (primary group), 40% (salvage group).	N/A.
[11]	2020	Prospective observational trial.	N/A.	Hemigland.	Yes: 3, 6, 18 months.	MRI.	61	Primitive.	N/A	No CsPCa at 6 months MRGB for 82%.	No Clavien dindo > 2.	82% RFS at 18 months.	N/A.
[12]	2020	Retrospective cohort study.	2b.	Combination.	Sys and target.	mpMRI.	Hemi: 26.	Primitive.	Urinary incontinence 17% (1 year) in both groups; impotency 75% (WGC) 46% (HC)	4y FFS 73%.	N/A.	N/A.	N/A.
[13]	2023	Retrospective cohort study.	All grades.	Focal.	PSA.	MRI.	163	Recurrence.	Urinary incontinence 1.8%; erectile dysfunction in 3.1% of patients.	BRFS 78%, 74%, and 55% for low, intermediate, and high-grade cancers.	N/A.	N/A.	N/A.
[14]	2023	Retrospective cohort study.	All grades.	Focal and hemi.	30 days, then every 3 months for the first 2 years, every 6 months from the third to the 50th year, and once a year until the 10th year.		110	Primitive.	N/A.	BCS and TFS of 68.5% and 71.5%.	N/A.	N/A.	N/A.
[15]	2023	Retrospective cohort study.	N/A.	Focal/hemi.	Biopsy at 2 year	MRI at 2 years.	132	Primitive.	N/A.	TFS in-field and out-of-field: 97% and 86%.	N/A.	TFS in-field and out-of-field: 97% and 86%.	N/A.
[16]	2023	Prospective cohort study	N/A.	Salvage focal ablation.	Biopsy	mpMRI.	7	Recurrence.	Erectile function was preserved.	5/7 disease-free at the most recent MRI control.	N/A.	5/7 disease-free at the most recent MRI control.	N/A.
[17]	2023	Retrospective cohort study.	N/A.	SWGC.	N/A.	N/A.	110	Primitive.	IIEF post cryoablation 1. Stress urinary incontinence post cryoablation (2%).	BRFS, FFS, and MFS at 10 years were 84%, 66%, and 96%, respectively.	Grade > 2 Clavien–Dindo adverse events: (2.3%) patients.	BRFS, FFS, and MFS at 10 years were 84%, 66%, and 96%, respectively.	N/A.
[18]	2021	Retrospective cohort study.	All grades.	WGC.	N/A.	N/A.	sCT = 186; sHIFU = 113; NST = 982.	Recurrence.	N/A.	N/A.	sCT: rectourethral fistulas (3%) and severe incontinence (7%).	CSS (*p* < 0.001) for CT.	OS (*p* < 0.001) for CT.
[19]	2021	Retrospective cohort study.	All grades.	WGC.	N/A.	MRI every 6–12 months.	108	Recurrence.	Group A: better clinical relief of urinary symptoms.	Reduced the risk of FFS by 45.8%.	Clavien—dindo Grade I: 13 (24.1%).	Reduced the risk of FFS by 45.8%.	No difference in the 2 groups.
[20]	2021	Prospective controlled trial.	GG 1–2.	Ipsilateral hemigland and contralateral anterior prostate.	6, 18, 36 months.	23 pt 6 month, 16 pt 18 months, 12 pt 36 months.	23	Primitive.	Sexual improvement after 6 months; 52% preserved urinary continence.	8/23 (34.8%) positive out of field biopsy within 3 years.	N/A.	N/A.	N/A.
[21]	2023	Pilot study design.	Gleason 6 and 7.	Focal.	Day 1 with cryoablation.	3 months and 1 year post-operative MRI.	N/A.	Primitive.	IIEF-5: 18/I-PSS score: 9.	At 3 months complete ablation index lesion, no signs of recurrence at 1 year.	N/A.	N/A.	N/A.
**Irreversible Electroporation**													
[22]	2021	Prospective cohort study.	GG 1–2.	IRE.	Biopsy (TTMB) at 12 mo.	MRI 12 months.	50	Primitive.	EPIC urinary or bowel QoL domain, decline in EPIC sexual QoL.	2.5% residual disease at 12 mo.	No Clavien–Dindo grade 3 events or higher.	N/A.	N/A.
[23]	2023	Retrospective cohort study.	Intermediate–high risk.	IRE localized cancer.	12 months biopsy.	MRI 6 months.	229	Primitive.	Erections sufficient for intercourse (71 to 58). Short-term urinary continence was preserved (99% 12 mo).	Kaplan–Meier FFS: 91% at 3 years, 84% at 5 years and 69% at 8 years.	N/A.	N/A.	PCa specific and overall survival were 100%.
[24]	2023	Prospective cohort study.	Low–intermediate.	IRE focal and extended.	6 months.	N/A.	106 (51 focal, 55 extended).	Primitive.	IIEF score and EPIC score was better in the focal group.	Rate of residual prostate cancer without significant difference in the 2 groups.	N/A.	N/A.	N/A.
[25]	2023	Retrospective cohort study.	ISUP 1–3 grade.	IRE focal.	12 months biopsy.	MRI 6 months.	45	Both.	Quality of life (QoL) no significant changes; mild decrease in sexual QoL.	FFS at 3 years was 96.75%, metastasis free survival in 99% and overall survival 100%.	No Clavien–Dindo > 1 complications were reported.	FFS 3 years 91.3%.	OS 3 years 100%.
[26]	2023	Retrospective cohort study.	ISUP 1–2 grade.	IRE focal.	12 months biopsy.	N/A.	41	Primitive.	All patients preserved urinary continence. Potency was maintained in 91.8%.	Recurrence was observed in 16 of 41 (39%) of the whole cohort.	N/A.	Median recurrence-free survival: 32 months (95% CI: 6.7–57.2).	N/A.
[27]	2022	Nonrandomized controlled trial—retrospective.	T2c or low, Gleason 7 or less.	Extended focal H-FIRE	6 months biopsy.	MRI 1 and 6 months.	109	Primitive.	International Prostate Symptom Score was 4.5. International Index of Erectile Function 5 score was 2.0.	csPCa AT 6 mo 6.0% (95% CI, 2.2%–12.6%; *p* < 0.001; 1 in the treatment zone and 5 outside the treatment zone).	Clavien–Dindo grade I (33 cases), II (7 cases), III (1).	N/A.	N/A.
[28]	2021	Retrospective cohort study.	T2c or low.	Focal IRE	N/A.	6 months MRI.	10	Primitive.	IIEF no significant changes; no new urinary incontinence developed.	9/10 reduction of diffuse restriction at 6 months and PIRADS decrease.	N/A.	N/A.	N/A.
**Microwave Ablation**													
[29]	2022	Single-center, prospective, interventional phase 1–2 trial.	Grade group 1 and 2.	Focal lesion TMA.	PSA, IPSS, and IIEF5 1 w, 1, 3, 6 mo. Rebiopsy 6 mo.	MRI 5 mo.	11	Primitive.	IPSS (*p* = 0.39), or IIEF-5 scores (*p* = 0.18), no significant changes.	Necrosis of the index tumor on MRI 8/10.	No grade ≥ 2 complications were reported.	N/A.	N/A.
[30]	2021	Prospective cohort study.	Gleason score ≤ 3 + 4.	Focal microwave ablation.	Rebiopsy 6 mo.	MRI 7 days.	10	Primitive.	No significant change of median IPSS, IIEF-5, and MSHQ-EjD at 6-month	Total necrosis of the index tumor on MRI 8/10 at 7 days.	N/A.	N/A.	N/A.
[31]	2022	Single center prospective phase 2 trial.	ISUP grade 2.	Focal microwave ablation.	Biopsy 5 months.	MRI 5 months.	15 men, 23 areas.	Primitive.	urinary symptoms, uroflowmetry, erectile function, and QOL scores. No significant difference at 6 months.	At 6 months 91.3% (21/23) no cancer; per-patient analysis 33.3% (5/15) positive.	Grade 1 complications: hematuria (33.3%), dysuria (6.7%), and perineal discomfort (13.4%).	N/A.	N/A.

## Data Availability

The authors confirm that the data supporting the findings of this study are available within the article and its Appendix A.

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
