# Peer review of "Focal Minimally Invasive Treatment in Localized Prostate Cancer: Comprehensive Review of Different Possible Strategies"

_cancers, 2024, doi:10.3390/cancers16040765_

Round 1

Reviewer 1 Report

Comments and Suggestions for Authors

Faiella et al have performed a comprehensive review of studies of cryoablation, electroporation and microwave ablation for prostate cancer from 2020-2023.  They identified 23 studies, the majority of which were retrospective.  My conclusion from this analysis is that there is very low quality of data to inform the efficacy and safety of local ablative techniques, due to the lack of prospective randomized or even non-randomized studies that compare these strategies to active surveillance or radical therapy.  The authors conclude that these ablative techniques are promising for "improved functional outcomes and reduce[d] adverse events." 

Specific comments:

1. The conclusions are not supported by the data presented.  the data is low quality.  The category of evidence for nearly all the identified studies is low (level 4 data, with just a couple prospective controlled nonrandomized studies that would be level 3).  All conclusions claiming improvement in outcomes function or oncologic are not appropriate.  Future study is warranted, but whether these interventions compared to active surveillance or radical therapy provide superior results either oncologically or adverse event wise is currently completely unexplored.

2. Simple summary Line 14-15: "Minimally invasive methods have demonstrated optimal functional outcomes and low adverse events." is not supported by evidence.

3. Clarify in abstract-- line 22 states that aim of review was to look at patients with locally advanced prostate cancer, but the studies did not necessarily only focus on this

4. Abstract line 33-34 "despite strong evidence that..." is not supported by evidence.

5. Meaning of "viger" page 2 line 48?

6. Page 2 line 6: plural consensus meetings is discussed but only one citation.  Cite addition if using pleural meetings vs a consensus meeting (singular that is cited).

7. Tables all formatted so that they are very difficult to understand.  Need to include the citations in the tables.  Include NCT# where appropriate for the prospective studies.  Given that this is a comprehensive review, assigning a level of evidence/quality rating and listed on the Tables would be helpful.

8. Page 7 line 95 unclear meaning of "started a year earlier"

9. Page 8 line 107-108: claim mendez found a significant different in what?  What was being compared in that sentence is unclear.

10. Page 8 line 123: for Gregg study, need to clarify that it is a prospective non-randomized controlled trial.  The use of the term "controlled" needs to be a very careful given that there were no control groups in any study presented.

11. Page 12 line 321: Citation needed for statement about comparison immune effects.

12. Discussion paragraphs 1-4: this information seems to all be introduction and not discussion.  Would integrate with the introduction and remove here.

13. Page 12 line 356: use of the term that studies "confirmed" is not appropriate given the level of evidence.

14. Page 13 line 411-413: it has not been demonstrated that these techniques are safe and effective due to lack of prospective randomized data.

15. Page 14 line 432: meaning of "resume"

16. Page 14 conclusion: the statement that effectiveness of minimally invasive treatment in comparison to standard of care is "strong supported" is incorrect.  no data is presented to compare to standard of care.

Comments on the Quality of English Language

just minor editing required.

Author Response

REVIEWER 1 REPLY

Faiella et al have performed a comprehensive review of studies of cryoablation, electroporation and microwave ablation for prostate cancer from 2020-2023. They identified 23 studies, the majority of which were retrospective. My conclusion from this analysis is that there is very low quality of data to inform the efficacy and safety of local ablative techniques, due to the lack of prospective randomized or even non-randomized studies that compare these strategies to active surveillance or radical therapy. The authors conclude that these ablative techniques are promising for "improved functional outcomes and reduce[d] adverse events."

Thank you for your considerations. Based on them we consider to introduce another paraghraph in the discussion section to underline the lack of prospective studies.

We added: “Evenmore, in the current literature there is a lack of prospective randomized and non-randomized studies that compare these strategies to active surveillance or radical therapy, leading to primordial results which need confirmation and validation in wider samples and multicentric evaluation.”

Specific comments:

  1. The conclusions are not supported by the data presented. the data is low quality. The category of evidence for nearly all the identified studies is low (level 4 data, with just a couple prospective controlled nonrandomized studies that would be level 3). All conclusions claiming improvement in outcomes function or oncologic are not appropriate. Future study is warranted, but whether these interventions compared to active surveillance or radical therapy provide superior results either oncologically or adverse event wise is currently completely unexplored.

Thank you again for these suggestions. We agree there is not strong evidence of the effectiveness of focal therapy by still promising results in terms of functional and adverse effects reduction. In addition to the before mentioned sentence we added in the conclusion the need for more comprehensive researches to fully understand and demonstrate the full oncological potential of minimally invasive techniques.

  1. Simple summary Line 14-15: "Minimally invasive methods have demonstrated optimal functional outcomes and low adverse events." is not supported by evidence.

Thank you for stressing out the “not so strong evidence” point. We fixed the issue: “Minimally invasive methods have demonstrated encouraging results in terms of  functional outcomes and low adverse events.”

  1. Clarify in abstract-- line 22 states that aim of review was to look at patients with locally advanced prostate cancer, but the studies did not necessarily only focus on this

Thank you, you are right since we didn’t just focus on locally advance prostate cancer, we fixed the issue and generalized the sentence reporting just “patients with prostate cancer”.

  1. Abstract line 33-34 "despite strong evidence that..." is not supported by evidence.

Thank you for highlighting the “not so strong evidence” point. We fixed the issue: “The oncological effectiveness of minimally invasive treatment in comparison to standard of care is still under investigation, despite encouraging results in terms of functional outcomes improvement and adverse events reduction.”

  1. Meaning of "viger" page 2 line 48?

Thank you for finding the typing error. The word was “higher”.

  1. Page 2 line 6: plural consensus meetings is discussed but only one citation. Cite addition if using pleural meetings vs a consensus meeting (singular that is cited).

Thank you for your suggestion. We specified the consensus meeting with the following sentence “KJ Tay et al. at the International Delphi Consensus meeting concluded that AS should be prioritized in males with low-risk illness because there is no net benefit from FT, whereas FT should be explored in individuals with intermediate PCa risk.”

  1. Tables all formatted so that they are very difficult to understand. Need to include the citations in the tables. Include NCT# where appropriate for the prospective studies. Given that this is a comprehensive review, assigning a level of evidence/quality rating and listed on the Tables would be helpful

Thank you for your corrections. We added the NCT for prospective studies when possible (some of them are not open access) and the citations inside the table. Because most of the studies are small sample studies we can assume they still don’t have significant levels of evidence.

Molti studi primordiali per cui non fondamentali i livelli di evidenza.

  1. Page 7 line 95 unclear meaning of "started a year earlier"

Thank you for your suggestion, we have tried to make the sentence more clear modifying it as it follows “the authors continue the cryoablation evaluation started by their previous study, still directed by Tan WP”.

  1. Page 8 line 107-108: claim mendez found a significant different in what? What was being compared in that sentence is unclear.

Thank you for your suggestion, we have tried to make the sentence more clear.

  1. Page 8 line 123: for Gregg study, need to clarify that it is a prospective non-randomized controlled trial. The use of the term "controlled" needs to be a very careful given that there were no control groups in any study presented.

Thank you a lot, as you suggested we fixed the issue adding “non-randomized”.

  1. Page 12 line 321: Citation needed for statement about comparison immune effects.

Thank you for the useful consideration, we fixed the issue removing the immune effects since as you highlighted was not sufficiently supported by research.

  1. Discussion paragraphs 1-4: this information seems to all be introduction and not discussion. Would integrate with the introduction and remove here.

Thank you for noticing the repetition, we removed the first 4 sentences to avoid repetition.

  1. Page 12 line 356: use of the term that studies "confirmed" is not appropriate given the level of evidence.

Thank you as you suggested we modified it since the level of evidence is not high enough.

  1. Page 13 line 411-413: it has not been demonstrated that these techniques are safe and effective due to lack of prospective randomized data.

Thank you for your suggestion, we fixed the issue.

  1. Page 14 line 432: meaning of "resume"

Thanks for pointing out it was not very clear. We cancelled it, since it was not necessary for the meaning of the sentence.

  1. Page 14 conclusion: the statement that effectiveness of minimally invasive treatment in comparison to standard of care is "strong supported" is incorrect. no data is presented to compare to standard of care.

Thank you for pointing this out. We fixed the issue.

Reviewer 2 Report

Comments and Suggestions for Authors

This study was reported the efficacy of focal therapy for low-risk prostate cancer. However, the reviewer would like to suggest some critiques to make this paper as follows.

Major revision

1.     On line 18, what is “selective treatment”? selective patients?

2.     On line 53, the reviewer thinks that the index lesion is not associated with local recurrence, distant metastasis, or oncologic outcome. Tumor size, position, and grade may be prognostic factors in prostate cancer and should be cited in the literature. The statement that "most concentrated cancer" is related to oncologic outcomes may be confusing to readers.

3.     After line 86, there are too many line breaks, making it very difficult for the reader to read. The paragraphs should be re-structured by category.

4.     On 316, the authors should delete this sentence as it is repetitive.

5.     Because prostate cancer is a multifocal tumor, it is impossible to detect all lesions by TRUS or MRI. Therefore, this should be mentioned in the limitation.

6.     FT is therefore a treatment for a limited number of patients. It should also be mentioned that the number of enrolled patients in the cited article is relatively small, although it has already been mentioned in the limitation.

Author Response

REVIEWER 2 REPLY

This study was reported the efficacy of focal therapy for low-risk prostate cancer. However, the reviewer would like to suggest some critiques to make this paper as follows.

Major revision

  1. On line 18, what is “selective treatment”? selective patients?

Thanks for casting the doubt, we modified the sentence to make it more clear “Focal therapy is a promising, minimally invasive method for the treatment of patients with localized prostate cancer”.

  1. On line 53, the reviewer thinks that the index lesion is not associated with local recurrence, distant metastasis, or oncologic outcome. Tumor size, position, and grade may be prognostic factors in prostate cancer and should be cited in the literature. The statement that "most concentrated cancer" is related to oncologic outcomes may be confusing to readers.

Thank you, as suggested we modified the sentence to make it more clear and avoid the reader confusion “FT aims to treat the “index lesion, saving adjacent critical structures. Tumor size, position and grade may determine the likelihood of metastases and consequently the prognosis of the patient.”

  1. After line 86, there are too many line breaks, making it very difficult for the reader to read. The paragraphs should be re-structured by category.

Thanks for this suggestion, we propose paraghraphs divided by treatment category (cryoablation, MWA and IRE). Please let us know if you have any other consideration.

  1. On 316, the authors should delete this sentence as it is repetitive.

Thanks for this consideration. We fixed the issue, removing the repeated sentence.

  1. Because prostate cancer is a multifocal tumor, it is impossible to detect all lesions by TRUS or MRI. Therefore, this should be mentioned in the limitation.

Good point, thank you for raising it. We added this revision to the limitations.

  1. FT is therefore a treatment for a limited number of patients. It should also be mentioned that the number of enrolled patients in the cited article is relatively small, although it has already been mentioned in the limitation.

Thank you for your suggestion. We fixed it out highlighting the low number of patients in papers sample as one of the main limitations.

Round 2

Reviewer 1 Report

Comments and Suggestions for Authors

Outstanding issues:

1. Remove "in comparison to standard of care" from simple summary (Page 1 Line 17.  The main limitation of this entire body of literature is lack of comparisons with standard of care.

2. Change statement in abstract Line 27-28 that "Paper that were prospective development stage 2a in design made up the majority."  23 studies are presented.  13 are retrospective.  The majority of studies are retrospective.  A stage of study is not appropriately assigned to retrospective studies.  This sentence needs to be accurate.  There is a similar statement on Page 2 line 83 in Results "The majority of studies were phase 2a retrospective cohort studies."  Need to remove phase 2a from this statement.

3. Table Wang N 2021: Group A vs B is not defined.  The discussion of this study Page 9 193-201 is difficult to understand given that groups are not defined.  The difference in symptoms doesn't appear to be discuss in text.

4. Conclusions: Page 14 line 257-259: The statement "Evenmore, the oncological effectiveness of minimally invasive treatment in comparison to standard of care, active surveillance or radical therapy, is supposed by evidence that it improves functional outcomes and reduces adverse events."  needs to be removed.  There is essentially no data presented that compares functional outcomes or adverse events with the standard of care.

5. Include the citations of the papers in the Table (not just the Name of author but the number as cited).

6. Include the NCT# (clinicaltrials.gov #) for registered trials in the Table.  If not applicable, this needs to be indicated in the Table as well.

Comments on the Quality of English Language

Some editing still required.

Author Response

REVIEWER 1

Outstanding issues:

  1. Remove "in comparison to standard of care" from simple summary (Page 1 Line 17).  The main limitation of this entire body of literature is lack of comparisons with standard of care.

Thank you for your correction. We removed the sentence.

  1. Change statement in abstract Line 27-28 that "Paper that were prospective development stage 2a in design made up the majority."  23 studies are presented.  13 are retrospective.  The majority of studies are retrospective.  A stage of study is not appropriately assigned to retrospective studies.  This sentence needs to be accurate.  There is a similar statement on Page 2 line 83 in Results "The majority of studies were phase 2a retrospective cohort studies."  Need to remove phase 2a from this statement.

Thank you very much for helping us fixing this error. We changed the sentence in the abstract results and removed “phase 2a” from the results sentence: “The majority of studies were retrospective cohort studies”.

  1. Table Wang N 2021: Group A vs B is not defined.  The discussion of this study Page 9 193-201 is difficult to understand given that groups are not defined. The difference in symptoms doesn't appear to be discuss in text.

Thank you for the observation. The Tan WP study you are referring to at lines 193-201 is the study of 2023 that does not have 2 groups but just 1 group undergoing Whole-gland Cryoablation for Radiation-resistant Prostate Cancer. Tan WP study of 2021 (the one with the 2 groups) is discussed at lines 113-132 (as written in the text group A underwent salvage partial gland ablation and group B underwent primary customized partial gland cryoablation, the 2 groups were compared for free-survival, incontinence and erectile function). We specified the two groups following your suggestion.

  1. Conclusions: Page 14 line 257-259: The statement "Evenmore, the oncological effectiveness of minimally invasive treatment in comparison to standard of care, active surveillance or radical therapy, is supposed by evidence that it improves functional outcomes and reduces adverse events."  needs to be removed.  There is essentially no data presented that compares functional outcomes or adverse events with the standard of care.

Thank you for your observation, we removed the sentence as requested and modified the conclusion paragraph: “The oncological effectiveness of minimally invasive treatment in comparison to standard of care is still under investigation, despite encouraging results in terms of functional outcomes improvement and adverse events reduction.”

  1. Include the citations of the papers in the Table (not just the Name of author but the number as cited).

Thank you for the suggestion, we fixed the issue inserting the number cited in the review next to the author’s name to avoid adding another column to a table that already has 14 columns.

  1. Include the NCT# (clinicaltrials.gov#) for registered trials in the Table.  If not applicable, this needs to be indicated in the Table as well.

Thank you for your correction. We already included the NCT for open access registered trial next to the study design (we did it this way in order to avoid adding another column to a table that already has 14 columns). For the rest of them when we could not access the NCT, we fixed the issue inserting “non applicable” in the table.

Reviewer 2 Report

Comments and Suggestions for Authors

none.

Author Response

thank you